# Internal Tide loss of coherence in a realistic simulation of the North Atlantic

Adrien Bella[1], Noé Lahaye[1], and Gilles Tissot[1]

[1]Inria, Odyssey team and IRMAR, Université de Rennes

**Correspondence:** Adrien Bella (adrien.bella@outlook.fr)

**Abstract.** The loss of coherence of the semidiurnal internal tide is investigated using a high-resolution realistic numerical simulation over the North Atlantic. The analysis focuses on processes resulting from the interaction between the internal tide and the mesoscale background flow at time scales typically shorter than one month. To this end, a theoretical framework based on vertical mode decomposition and the splitting of the internal tide signal into coherent and incoherent components is developed and applied to the outputs of the numerical simulation. This framework enables the transfer terms between the coherent and incoherent parts, and between the different vertical modes — and therefore horizontal scales — of the internal tides to be evaluated. By focusing on three subdomains with contrasting dynamics, we demonstrate that coherent-to-incoherent energy transfers significantly impact the internal tide energy budget. These transfers are dominated by advection by slowly varying flows and mainly occur without changing the vertical mode of the internal tide involved. This is attributed to the dominance of the barotropic and first baroclinic modes in the mesoscale flow combined with the structure of the mesoscale flow/internal tide interaction terms. Typical energy transfer rates are of the order of a few tens of days in the Gulf Stream region and a few hundred days in the Azores for the mode 1 internal tide.

## 1 Introduction

Internal tides are a major component of the internal wave field in the ocean. They play an important role in the energy transfer in the ocean, including dissipation and mixing routes, and in shaping the global circulation (*e.g.* Whalen et al., 2020; Jayne et al., 2004; Wunsch and Ferrari, 2004). They also present a major observational challenge, particularly in the context of the recently launched SWOT mission (Morrow et al., 2019), as their signature entangles with the submesoscale dynamics (Arbic et al., 2015; Torres et al., 2019).

As they propagate through the unsteady ocean, internal tides (IT) lose their fixed-phase relationship with the astronomical forcing – a process often referred to as loss of coherency, and which will be referred to as such in this paper. In principle, incoherent internal tides can also be caused by the barotropic tide being already incoherent, although this is mostly not the case – at least in the deep ocean (*e.g.* Kelly et al., 2015; Shriver et al., 2014). The incoherence of IT poses a significant challenge

to quantifying the IT field using satellite observations. This is because the coarse sampling of the data necessitates the use of long time series to extract the tidal signal, which only provides access to its coherent part. Incoherent internal tides were identified decades ago (*e.g.* Munk et al., 1965; Munk and Cartwright, 1966; Colosi and Munk, 2006) and have been measured based on various types of observations eversince. The reader is referred to the introductions of Ponte and Klein (2015) or Buijsman et al. (2017), among others, for more exhaustive reviews of the literature on the subject. It is currently accepted that, on average, more than half of the internal tide variance is incoherent, as evidenced by satellite altimeter data (Zaron, 2017), ARGO parking-phase data (Geoffroy and Nycander, 2022) or realistic numerical simulations (Nelson et al., 2019; Lahaye et al., 2024).

To which extent the internal tide signal is incoherent has thus been widely documented by means of signal processing, including the use of in situ observations (*e.g.* Nash et al., 2012; Kelly et al., 2015). Likewise, the mechanisms associated with the loss of coherence of internal tides have been identified and discussed for a long time (*e.g.* Kunze, 1985; Rainville and Pinkel, 2006; Zaron and Egbert, 2014). Generally speaking, loss of coherence is the direct consequence of the time in-homogeneity of the medium of propagation of a wave. As summarised in Savage et al. (2020), and besides the possibility of the barotropic tide to be already incoherent (Bendinger et al., 2025), the internal tide becomes incoherent by interacting with the background currents (mostly via advection) and/or via refraction, which can be due to fluctuations of the wave horizontal propagation velocity associated with variations of the background stratification profile. The loss of coherence due to advection by the mean flow can be described as follows: at leading order, (*i.e.* when the wave field can be described as a superposition of local plane waves), the advection term can be interpreted as a transport of the wave by the mean flow. This results in a local phase perturbation that propagates afterwards. As this process is not constant over time because the mesoscale flow evolves, the wave field becomes randomly perturbed and hence incoherent. Although these processes are well understood in principle, quantitative analyses – by means of theory, observations and numerical simulations – are still needed. In particular, diagnostics based on the dynamical equations, allowing to validate and provide a more quantitative understanding of these dynamical mechanisms, remain rare in the literature (with the notable exception of Savage et al. (2020) in the Tasman Sea).

In this paper, we analyse the interaction terms between the low-frequency (mostly mesoscale) flow component and the semidiurnal internal tide which result in a loss of coherence of the latter, using outputs from a high-resolution realistic numerical simulation of the North Atlantic ocean. We mostly use the same data and analysis framework as in Bella et al. (2024), hereafter referred to as **Ba24**, although the present study is restricted to a few domains of interest and uses an extended framework to discuss energy exchanges between the coherent and incoherent internal tide, and the loss of internal tide coherence. Our methodology is based on a vertical mode decomposition of the linearised equations (around a low-frequency flow), which are further separated into a coherent (harmonic) and incoherent part, and from which the energy budget is constructed. The corresponding terms are then evaluated from the simulation outputs. We focus on loss of coherence occurring over timescales of a month or less, thus addressing interactions between the internal tide field and the mesoscale eddy field. The impact of variability at lower frequency (e.g. seasonal changes in the background stratification) is not investigated here.

The paper is organised as follows. The next section 2 introduces the numerical simulation, the data processing as well as the theoretical framework used to conduct this study. Results are presented and discussed in Section 3, and the conclusions are given in Section 4.

## 2 Data and methods

This study is based on the analysis of outputs from a high-resolution numerical simulation of the North Atlantic ocean. We use a theoretical framework based on linear theory of the internal tides, which are decomposed into vertical modes as well as into coherent and incoherent contributions. Apart from the coherent/incoherent separation, the data and analysis are very similar to the one presented in **Ba24** (and also, to some extent, in Lahaye et al. (2024)). Therefore, only the key points are provided below and readers are referred to the two aforementioned papers for further details.

### 2.1 Description of the eNATL60 numerical simulation

#### 2.1.1 Model configuration and validation

The numerical simulation eNATL60 (Brodeau et al., 2020) is a realistic simulation of the North Atlantic Ocean based on the NEMO model (Nucleus for European Modelling of the Ocean, Madec et al., 2019). It has a 1/60° horizontal grid resolution (around $1.5\,\mathrm{km}$ at mid-latitude) and 300 "partial steps" vertical levels (*i.e.* fixed vertical levels, except near the seafloor where it is modified to match the local depth – see Madec et al. (2019)). Imposed external forcing are taken from the ERA-analysis for surface atmospheric forcing, and the FES2014 atlas (Lyard et al., 2020) for the barotropic tidal forcing at the boundaries as well as the corresponding tidal potential within the domain. The tidal constituents $M_2, S_2, N_2, K_1$ and $O_1$ were included. The numerical simulation ran for 13 months, after an 18-month spin-up period during which tidal forcing was activated for the final six months. The simulation was initialised from a 1/12° reanalysis (GLORYS12v1). To date, this simulation is one of the few realistic numerical simulations that achieve "submesoscale-permitting" resolution and explicitly include barotropic tidal forcing – and thus an internal tide field – while covering an entire basin. The domain of the numerical simulation is shown in Figure 1.

Validation of the eNATL60 simulation has been addressed in different studies, and the main results are summarised here. Brodeau et al. (2020) provided various validation materials. Notably, a comparison of the barotropic tide with the FES2014 tidal atlas, which is used for boundary tidal forcing in the simulation, revealed good agreement, particularly with respect to the dominant semi-diurnal amplitude. Furthermore, a comparison of the mesoscale field with the AVISO/DUACS product (*i.e.* comparing the standard deviation of daily-averaged SSH at a similar spatial resolution) revealed a reasonable degree of agreement. Yet, as expected given the coarser resolution of the AVISO/DUACS product, eNATL60 is more energetic. Further intercomparison of submesoscale-permitting numerical simulations by Uchida et al. (2022) shows that eNATL60 falls within the range of various models in terms of mesoscale energy and dominant patterns (*e.g.* the mean location of the Gulf Stream). Comparisons of semidiurnal energy with drifter data (from Caspar-Cohen et al., 2025), and of coherent IT (although the time

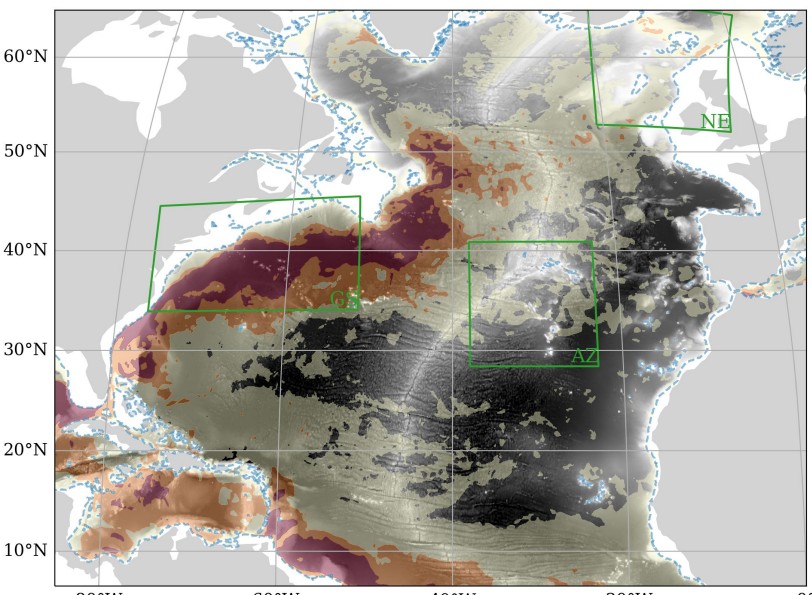
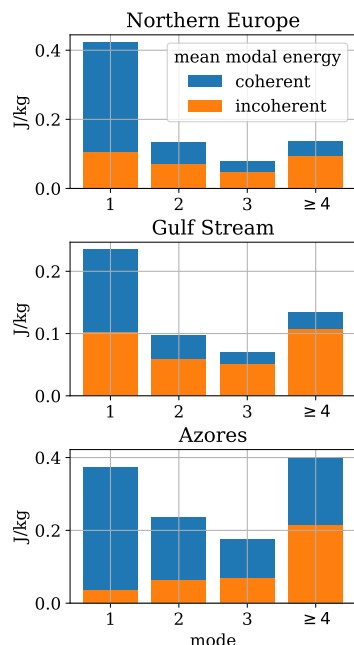

**Figure 1.** (left) a portion of the domain of the eNATL60 simulation, showing the bathymetry (black shading) and the standard deviation of the low frequency surface currents (filtered at 2 days) computed over 1 year (yellow-to-red, contours at $15\,\mathrm{cm/s}$, $30\,\mathrm{cm/s}$ and $45\,\mathrm{cm/s}$). The contour of the mask based on the minimum depth ($h > 250\,\mathrm{m}$) is plotted in dashed blue, and the three subdomains of interest are delimited with green lines: NE: Northern Europe; GS: Gulf Stream and AZ: Azores. (right) incoherent and coherent (computed over 1 month) mean modal energy over each subdomain (averaged over the 4 months analysed).

window for its definition is inevitably different) with estimates from satellite (HRET, Zaron, 2019), is available in Lahaye et al. (2024, see their Supporting Information). It shows that the main beams are captured in eNATL60 (compared to HRET),
although the amplitude is larger – as expected given the shorter time window for computing the coherent signal using harmonic analysis. Conversely, the surface semidiurnal horizontal kinetic energy agrees with the drifter-derived estimate within a factor of $0.5$ to $1.5$ across most of the domain.

### 2.1.2   Description of the data used for the analysis

We have analysed 8 months of hourly data (pressure and horizontal velocity), from July 2009 to February 2010, that have been
projected onto the vertical modes (see **Ba24**, Lahaye et al. (2024) and the section below for the definition of the vertical modes). The present analysis, including the decomposition into coherent and incoherent contributions, relies mostly on 4 months covering late summer to winter, namely August, October, December and February. These four month should be representative of the conditions one can encounter in the North Atlantic, and allow to sample over seasonal variations, albeit not exhaustively. Furthermore (and mostly for matters of computational cost), our study focuses on three subdomains of interest (see Fig. 1). These

three regions were shown (Bella et al., 2024) to feature distinct configurations of the internal tide and its interactions with the low-frequency circulation, as well as being representative of the broader eNATL60 domain: the first domain, located around the Azores, has strong IT generation that propagates through weak mesoscale currents; the second domain, around the Gulf Stream, has IT generation and propagation inside an energetic mesoscale field; and the last domain, in the north-eastern part of the basin, exhibits IT generation where waves cannot propagate far (because of the small horizontal propagation velocity) as well as active mesoscale currents. In **Ba24**, these three areas were found to be representative of three different dynamical regimes for the internal tide energy budget (5 other subdomains exhibited very similar properties): the Azores were dominated by interactions with the topography; the Gulf Stream region showed a forward energy cascade caused by the advection of the IT by the low frequency circulation; and the northern domain hosted energy exchanges between the low frequency flow and the internal tide through shear production terms.

## 2.2 Theoretical framework for diagnosing the internal tide propagation and loss of coherence

### 2.2.1 Vertical mode decomposition and Coupled Shallow Water equations

As mentioned previously, our theoretical framework is based on vertical mode decomposition, which has been derived and used in various studies in the context of internal tides dynamics in an inhomogeneous environment with a background flow (Kelly and Lermusiaux, 2016; Kelly et al., 2016; Bella et al., 2024).

Starting from the primitive equations with the hydrostatic and Boussinesq approximations, linearized around a background flow with low Rossby number (such that it can be assumed steady over one internal wave period), the variables are expanded over a series of vertical normal modes:

$$[p(\boldsymbol{x},z,t),\boldsymbol{u}(\boldsymbol{x},z,t)] = \sum_n [p_n(\boldsymbol{x},t),\boldsymbol{u}_n(\boldsymbol{x},t)]\phi_n(z;\boldsymbol{x}), \quad [w(\boldsymbol{x},z,t),b(\boldsymbol{x},t)] = \sum_n [w_n(\boldsymbol{x},t),b_n(\boldsymbol{x},t)N^2(z;\boldsymbol{x})]\Phi_n(z;\boldsymbol{x}), \quad (1)$$

where $p$ denotes the pressure, $\boldsymbol{v} = (\boldsymbol{u},w)$ the velocity (split in horizontal and vertical component) and $b$ the buoyancy. The vertical modes are defined at each horizontal position $\boldsymbol{x}$ following the standard Sturm-Liouville problem (*e.g.* Gill, 1982; Kelly, 2016), with a free surface:

$$\frac{d^2\Phi_n}{dz^2} + \frac{N^2}{c_n^2}\Phi_n = 0, \quad \Phi_n = \frac{c_n^2}{g}\frac{d\Phi_n}{dz}, z = \eta_m, \quad \Phi_n = 0, z = -H, \quad (2)$$

where $N^2(z;\boldsymbol{x})$ is the Brunt–Väisälä frequency, $c_n^2$ the eigenvalue which corresponds to the horizontal modal phase speed, and $\eta_m$ and $-H$ are the mean surface elevation and bottom depth, respectively. The vertical modes for the pressure and horizontal velocity are given by $\phi_n = \partial_z\Phi_n$ and are solutions of a sibling Sturm-Liouville problem.

Projection of the primitive equations onto the vertical modes yields a set of Coupled Shallow Water equations (Bella et al., 2024; Kelly and Lermusiaux, 2016). Neglecting some of the terms (compared to **Ba24**), because they were found to be of minor importance in the internal tide modal energy budget (in agreement with previous findings reported by Savage et al. (2017)) and

will not be investigated in this study – which focus on the impact of the background currents –, these equations are:

$$\partial_t \boldsymbol{u}_n + \sum_m \overline{\boldsymbol{U}}_{nm} \cdot \boldsymbol{\nabla} \boldsymbol{u}_m + \boldsymbol{f} \times \boldsymbol{u}_n + \boldsymbol{\nabla} p_n = -\sum_m \left[ \boldsymbol{T}_{nm} p_m + \boldsymbol{U}^h_{nm} \boldsymbol{u}_m + w_m \boldsymbol{U}^z_{nm} \right], \tag{3}$$

$$\partial_t p_n + \sum_m \overline{\boldsymbol{U}}^p_{nm} \cdot \boldsymbol{\nabla} p_m + \frac{c_n{}^2}{H} \boldsymbol{\nabla} \cdot (H \boldsymbol{u}_n) = \sum_m \left[ c_n^2 \boldsymbol{T}_{mn} \cdot \boldsymbol{u}_m + \boldsymbol{u}_m \cdot \boldsymbol{B}_{nm} \right]. \tag{4}$$

Here, $\boldsymbol{\nabla} = (\partial_x, \partial_y)$ denotes the gradient with respect to the horizontal coordinates only. The neglected terms include the direct forcing by the tidal potential (it is significant only for the barotropic mode while we focus on the baroclinic tide), horizontal gradient of the background stratification and perturbation of the stratification with respect to this background profile. We

recall that the background stratification $N^2$ is used to define the vertical modes, and a monthly average is considered for this definition. This time is short enough such that the perturbations around the mean stratification profile are small. Effects associated with the deviation of the free surface (*e.g.* horizontal gradient of the mean surface elevation) are small and, therefore, not included either. The various matrices and tensors entering this equation are:

$$\boldsymbol{T}_{nm} = \frac{1}{H} \int_{-H}^{\bar{\eta}} \phi_n \boldsymbol{\nabla} \phi_m \, dz, \qquad\qquad \overline{\boldsymbol{U}}_{nm} = \frac{1}{H} \int_{-H}^{\bar{\eta}} \boldsymbol{U} \phi_n \phi_m \, dz, \quad \overline{\boldsymbol{U}}^p_{nm} = \frac{1}{H} \int_{-H}^{\bar{\eta}} \boldsymbol{U} \frac{N^2}{c_m^2} \Phi_n \Phi_m \, dz,$$

$$\left( \boldsymbol{U}^h_{nm} \right)_{ij} = \frac{1}{H} \int_{-H}^{\bar{\eta}} \phi_n \phi_m \frac{\partial U_i}{\partial x_j} \, dz, \qquad \boldsymbol{U}^z_{nm} = \frac{1}{H} \int_{-H}^{\bar{\eta}} \phi_n \Phi_m \frac{\partial \boldsymbol{U}}{\partial z} \, dz, \quad \boldsymbol{B}_{nm} = \frac{1}{H} \int_{-H}^{\bar{\eta}} \Phi_n \phi_m \boldsymbol{\nabla}_h B \, dz.$$

From these equations, one can form the modal energy budget:

$$\frac{\partial E_n}{\partial t} + \boldsymbol{\nabla} \cdot \boldsymbol{F}_n = H \sum_m \left( \boldsymbol{C}_{nm} - \boldsymbol{A}_{nm} + \boldsymbol{H}_{nm} + \boldsymbol{V}_{nm} + \boldsymbol{P}_{nm} \right), \tag{5}$$

where $E_n$ and $\boldsymbol{F}_n$ are the modal vertically integrated energy (surface density) and energy flux:

$$E_n = H \left( \frac{\boldsymbol{u}_n^2}{2} + \frac{p_n^2}{2c_n{}^2} \right), \quad \boldsymbol{F}_n = H \boldsymbol{u}_n p_n,$$

and the right hand side terms are:

- $\boldsymbol{A}_{nm} = [\overline{\boldsymbol{U}}_{nm} \cdot \boldsymbol{\nabla} \boldsymbol{u}_m] \cdot \boldsymbol{u}_n + \overline{\boldsymbol{U}}^p_{nm} \cdot \boldsymbol{\nabla} p_m \, p_n/c_n^2$: advection by the mean flow;

- $\boldsymbol{C}_{nm} = \boldsymbol{u}_m \cdot \boldsymbol{T}_{mn} p_n - \boldsymbol{u}_n \cdot \boldsymbol{T}_{nm} p_m$: scattering term by the horizontal variations of the vertical mode basis, primarily associated with variations of the topography, but also by the background stratification;

- $\boldsymbol{H}_{nm} = -(\boldsymbol{U}^h_{nm} \boldsymbol{u}_m) \cdot \boldsymbol{u}_n$: horizontal shear production;

- $\boldsymbol{V}_{nm} = -w_m \boldsymbol{U}^z_{nm} \cdot \boldsymbol{u}_n$: vertical shear production;

- $\boldsymbol{P}_{nm} = \boldsymbol{u}_m \cdot \boldsymbol{B}_{nm} p_n/c_n^2$: buoyancy shear production.

### 2.2.2 Coherent/incoherent modal energy budget

For our analysis, we further decompose the modal internal tide field into a coherent and incoherent part and form a modal energy budget equation (this approach is similar to Savage et al. (2020)). The resulting energy budget will thus describe energy exchanges between the coherent and the incoherent tide. The coherent signal is defined as a sum over the semidiurnal astronomical frequency constituents that are included in the numerical simulation, and the incoherent part is the residual:

$$\boldsymbol{u}_n^c(t) = \mathcal{L}_c(\boldsymbol{u}_n(t)) = 2\Re\left[\sum_k \hat{\boldsymbol{u}}_n^k e^{i\omega_k t}\right], \quad \boldsymbol{u}_n^i(t) = \mathcal{L}_i(\boldsymbol{u}_n(t)) = \boldsymbol{u}_n(t) - \boldsymbol{u}_n^c(t) \tag{6}$$

(and likewise for $p_n$), with $k \in M_2, S_2, N_2$ and where $\hat{\boldsymbol{u}}_n^k$ denotes the harmonic amplitude at frequency $\omega_k$. We also introduced the coherent and incoherent extraction operators $\mathcal{L}_c$ and $\mathcal{L}_i = \mathbf{I} - \mathcal{L}_c$. Their numerical implementation will be described later on in the manuscript (eq. 17). Notice that we assume that the coherent and incoherent part are orthogonal with respect to the time average inner product (Savage et al., 2020; Wunsch, 2006), which will be valid in practice as the harmonic coefficients are determined by least square regressions (details are given in Appendix A), that is e.g.

$$\overline{\boldsymbol{u}^c p^i} = 0. \tag{7}$$

We checked that this property is indeed verified in our diagnostics.

Applying this operator to the CSW equations (3-4), one obtains the coherent CSW equations, and can readily form the incoherent equations by taking the residual:

$$\partial_t \boldsymbol{u}_n^c + \mathcal{L}_c\left[\sum_m \overline{\boldsymbol{U}}_{nm} \cdot \boldsymbol{\nabla} \boldsymbol{u}_m\right] + \boldsymbol{f} \times \boldsymbol{u}_n^c + \boldsymbol{\nabla} p_n^c = -\sum_m \left(\boldsymbol{T}_{nm} p_n^c + \mathcal{L}_c\left[\boldsymbol{U}_{nm}^h \boldsymbol{u}_m + w_n \boldsymbol{U}_{nm}^z\right]\right), \tag{8}$$

$$\partial_t p_n^c + \mathcal{L}_c\left[\sum_m \overline{\boldsymbol{U}}_{nm}^p \cdot \boldsymbol{\nabla} p_m\right] + \frac{c_n{}^2}{H}\boldsymbol{\nabla} \cdot (H\boldsymbol{u}_n^c) = \sum_m \left(\boldsymbol{T}_{mn} \cdot \boldsymbol{u}_m^c + \mathcal{L}_c\left[\boldsymbol{u}_m B_{nm}\right]\right), \tag{9}$$

$$\partial_t \boldsymbol{u}_n^i + \mathcal{L}_i\left[\sum_m \overline{\boldsymbol{U}}_{nm} \cdot \boldsymbol{\nabla} \boldsymbol{u}_m\right] + \boldsymbol{f} \times \boldsymbol{u}_n^i + \boldsymbol{\nabla} p_n^i = -\sum_m \left(\boldsymbol{T}_{nm} p_n^i + \mathcal{L}_i\left[\boldsymbol{U}_{nm}^h \boldsymbol{u}_m + w_n \boldsymbol{U}_{nm}^z\right]\right), \tag{10}$$

$$\partial_t p_n^i + \mathcal{L}_i\left[\sum_m \overline{\boldsymbol{U}}_{nm}^p \cdot \boldsymbol{\nabla} p_m\right] + \frac{c_n{}^2}{H}\boldsymbol{\nabla} \cdot (H\boldsymbol{u}_n^i) = \sum_m \left(\boldsymbol{T}_{mn} \cdot \boldsymbol{u}_m^i + \mathcal{L}_i\left[\boldsymbol{u}_m B_{nm}\right]\right). \tag{11}$$

One sees, in these equations, the terms associated with loss of coherence of the internal tide, which are identified as coupling terms between the coherent and incoherent sets of equations (terms with no "$\cdot^c$" or "$\cdot^i$" on the IT field variables). One can then derive a modal energy equation for the time-averaged coherent and incoherent energy, by taking the dot product of the momentum equations above with $\boldsymbol{u}_n$ and of the pressure by $p_n/c_n{}^2$. One thus obtains:

$$\boldsymbol{\nabla} \cdot \overline{\boldsymbol{F}_n^c} = H\sum_m \overline{\boldsymbol{C}_{nm}^c - \boldsymbol{A}_{nm}^{cc} - \boldsymbol{A}_{nm}^{ci} + \boldsymbol{H}_{nm}^{cc} + \boldsymbol{H}_{nm}^{ci} + \boldsymbol{V}_{nm}^{cc} + \boldsymbol{V}_{nm}^{ci} + \boldsymbol{P}_{nm}^{cc} + \boldsymbol{P}_{nm}^{ci}}, \tag{12}$$

$$\boldsymbol{\nabla} \cdot \overline{\boldsymbol{F}_n^i} = H\sum_m \overline{\boldsymbol{C}_{nm}^i - \boldsymbol{A}_{nm}^{ic} - \boldsymbol{A}_{nm}^{ii} + \boldsymbol{H}_{nm}^{ic} + \boldsymbol{H}_{nm}^{ii} + \boldsymbol{V}_{nm}^{ic} + \boldsymbol{V}_{nm}^{ii} + \boldsymbol{P}_{nm}^{ic} + \boldsymbol{P}_{nm}^{ii}}, \tag{13}$$

where $\boldsymbol{F}_n^c = H\boldsymbol{u}_n^c p_n^c$ and $\boldsymbol{F}_n^i = H\boldsymbol{u}_n^i p_n^i$ are the coherent and incoherent modal (horizontal) energy fluxes. The topographic scattering term is unchanged from before (but includes only coherent or incoherent contributions) and reads

$$\boldsymbol{C}_{nm}^r = p_n^r \boldsymbol{u}_m^r \cdot \boldsymbol{T}_{mn} - p_m^r \boldsymbol{u}_n^r \cdot \boldsymbol{T}_{nm}, \tag{14}$$

where $r$ is in the set $\{c, i\}$ (as will be $q$ below). The other terms are detailed below.

As could be expected from the coherent/incoherent CSW equations (8-11), the terms that are linear (time-derivative, Coriolis, topographic scattering) do not couple the coherent and incoherent parts. On the contrary, the terms that are associated with time-variable coefficients (such as low frequency current $U$ or low frequency buoyancy $B$) lead to a coupling between the coherent and the incoherent component. We further decomposed these terms into contributions that explicitly exhibit – or do not exhibit

– these interactions as follows:

- $\boldsymbol{A}_{nm}^{qr} = \mathcal{L}_q\left(\overline{\boldsymbol{U}}_{nm} \cdot \boldsymbol{\nabla}\boldsymbol{u}_m^r\right) \cdot \boldsymbol{u}_n^q + \mathcal{L}_q\left(\overline{\boldsymbol{U}}_{nm}^p \cdot \boldsymbol{\nabla}p_m^r\right)p_n^q/c_n^2;$

- $\boldsymbol{H}_{nm}^{qr} = -\mathcal{L}_q\left(\boldsymbol{U}_{nm}^h \boldsymbol{u}_m^r\right) \cdot \boldsymbol{u}_n^q;$

- $\boldsymbol{V}_{nm}^{qr} = -\mathcal{L}_q\left(w_n^r \boldsymbol{U}_{nm}^z\right) \cdot \boldsymbol{u}_n^q;$

- $\boldsymbol{P}_{nm}^{qr} = \mathcal{L}_q\left(\boldsymbol{u}_m^r \cdot \boldsymbol{\nabla}_h B_{nm}\right)p_n^q/c_n{}^2.$

Using this notation, equations (12) and (13) can be compactly rewritten as

$$\boldsymbol{\nabla} \cdot \overline{\boldsymbol{F}}_n^r = H \sum_{m,q}(\overline{\boldsymbol{C}_{nm}^r - \boldsymbol{A}_{nm}^{rq} + \boldsymbol{H}_{nm}^{rq} + \boldsymbol{V}_{nm}^{rq} + \boldsymbol{P}_{nm}^{rq}}). \tag{15}$$

### 2.2.3 Physical interpretation and separation into (anti)symmetric component

The various terms exhibited above couple the modes and coherent/incoherent contributions of the internal tide field with each others, in the sense that they describe exchange of energy between the vertical modes of the coherent and incoherent IT. In

general, when using a framework that decomposes a given field into different contributions, one may find useful to decompose the exchange terms into symmetric and antisymmetric parts. In particular, the antisymmetric part vanishes upon summation of the contributions, thereby describing exchanges that are not associated with a gain or loss of energy in the total field, while the symmetric part has a non-vanishing residual. This was used for instance in **Ba24** and Savage et al. (2020), where the modal exchange matrices where split into a symmetric and antisymmetric component as follows (say, for a matrix $\boldsymbol{Q}_{nm}$):

$\mathrm{Sym}(\boldsymbol{Q})_{nm} = (\boldsymbol{Q}_{nm} + \boldsymbol{Q}_{mn})/2, \quad \mathrm{Asy}(\boldsymbol{Q})_{nm} = (\boldsymbol{Q}_{nm} - \boldsymbol{Q}_{mn})/2.$

For the present analysis, we need to generalise this decomposition by performing the symmetric/antisymmetric decomposition on both the vertical mode and coherent/incoherent part:

$$\mathrm{Sym}(\boldsymbol{Q})_{nm}^{qr} = (\boldsymbol{Q}_{nm}^{qr} + \boldsymbol{Q}_{mn}^{rq})/2, \quad \mathrm{Asy}(\boldsymbol{Q})_{nm}^{qr} = (\boldsymbol{Q}_{nm}^{qr} - \boldsymbol{Q}_{mn}^{rq})/2. \tag{16}$$

The interpretation of this decomposition can be understood via the following considerations:

– Taking the sum over the coherent and incoherent parts ($r$ and $q$), one obtains the mode-wise symmetric and antisymmetric modal energy exchange matrices (from the symmetric and antisymmetric matrices, respectively) for the total (coherent + incoherent) IT modal energy equation;

– Conversely, taking the sum over the vertical modes ($n$ and $m$), one obtains the coherent/incoherent energy exchange terms for the tidal field;

– Taking the sum over both the vertical modes and the coherent/incoherent parts, the antisymmetric part of the matrix vanishes while the symmetric part gives a residual that denotes gain/loss of energy for the total tidal field;

– in any case, a given element of the matrix $Q_{nm}^{qr}$ represents energy transfers between the $q$ mode number $n$ and the $r$ mode number $m$, where $q$ and $r$ can be $c$ (for "coherent") or $i$ (for "incoherent").

Thus, in the following results, we will mostly focus on the antisymmetric part with $q = c, r = i$, which describes exchanges
between the coherent and incoherent parts.

## 2.3  Analysis of eNATL60 outputs

Our analysis consists of the following steps, which follows the theoretical framework described above. We projected 8 months of hourly outputs of pressure and horizontal velocity onto a set of 11 vertical modes, from $n = 0$ (barotropic mode) to $n = 10$ included, computed using the monthly-averaged background stratification. Sample tests show that the first 11 modes capture
more than 90% of the variance of pressure and horizontal velocity in the low-frequency band and the semidurnal band, in the vast majority of the basin (not shown). Exceptions are found in localized and patchy area for the horizontal velocity in the semidiurnal band (mostly in regions where the total semidiurnal variance is weak), where the relative variance residual peaks to 40%. In any case, the horizontal resolution of eNATL60 does not allow to resolve modes with $n >= 11$ everywhere in the North Atlantic. More precisely, as was described in **Ba24**, we computed the vertical mode projection using a 8-month
averaged vertical mode basis, and then re-project the modal amplitude on the monthly basis by using a cross-projection matrix between the 8-month average and monthly-average bases. This introduces a truncation error, which we found to be negligible for low-order modes (below 5).

From the time series of modal amplitudes, we ran a low-pass time filter to extract the mesoscale flow and complex demodulation to extract the semidiurnal tides. The same low-pass filter as for the mesoscale component was used in the complex
demodulation, for consistency. We used a fourth-order Butterworth filter with a time cutoff of 2 days. The choice of cutoff was motivated by the necessity to retain as much mesoscale variability as possible while discarding the diurnal tide and most of the near-inertial waves (the near-inertial period is two days at a latitude of $14°$, but all the subdomains included in our analysis are north of this latitude). The complex demodulation period is $12.2\,\mathrm{h}$, which is in the middle of the three semi-diurnal tidal components. The equivalent frequency band of the complex demodulation is $1.97 \pm 0.5\,\mathrm{cpd}$, which includes the spectral widening
associated with the incoherence of the internal tide signal.

From the complex-demodulated time series of modal amplitudes for the semidiurnal tide, we computed the coherent part using harmonic analysis based on least square fitting. For a given time series $f(t)$, and denoting its complex demodulated $\tilde{f}$ at frequency $\omega_c$, we thus computed:

$$\mathcal{L}_c f = 2\Re \left[ \sum_{k \in \{M_2, S_2, N_2\}} \hat{f}_k e^{i\omega_k t} \right], \quad \{\hat{f}_k\}_{k \in \{M_2, S_2, N_2\}} = \operatorname{argmin} \left| \tilde{f} - \sum_{k \in \{M_2, S_2, N_2\}} \hat{f}_k e^{i(\omega_k - \omega_c)t} \right|^2. \tag{17}$$

The different terms of the energy budget exposed earlier were then computed and averaged over each of the four months analysed. This allows to estimate the time variability of the obtained results. In addition, an estimate using an average (and definition of the coherent tide) over three months (September to November, included) was also performed. It qualitatively confirms the results obtained from the 1 month estimates (see discussion in Sec. 3.4). We discard regions shallower than 250 m, thus restraining this study to internal tides in the deep ocean (although the impact of continental shelves and islands is already noticeable cutting at this depth). Finally, most maps of computed fields that are shown in this paper are smoothed using a Gaussian filter with a kernel size of half the typical mode 1 lengthscale.

## 3    Results

### 3.1    Qualitative investigation in the Gulf Stream region

Let us first look at the coherent/incoherent energy budget in the Gulf Stream region, looking at the modal energy and flux divergence for the first two baroclinic modes over the month of October (Figure 2). The energy flux divergence $\boldsymbol{\nabla} \cdot \boldsymbol{F}_n^r$ indicates where (and to which amount) the coherent/incoherent internal tide is being generated or dissipated, which can be due to transfers with other (or same) coherent/incoherent modes or associated with net generation or dissipation.

In the Gulf stream domain, the corresponding patterns for mode 1 are relatively simple to interpret. ITs are mostly generated at the shelf break (red contours are visible in the Figure for both the coherent and incoherent parts), and subsequently propagate offshore. A zone of coherent flux convergence (loss) located around 40.5° N, 64° W, which is co-located with positive divergence for the incoherent part, clearly indicates loss of coherence of mode 1. The energy flux (arrows) flowing from the source (red color) to the sinks (blue color) also reflects this transfer from the coherent to incoherent component during its propagation (akin to Buijsman et al. (2017), see their Figure 4). Further offshore, the incoherent tide looses energy, and this loss is not compensated by a gain of energy of the coherent part, thus indicating energy transport (*e.g.* via advection by the mean flow) or transfer to *e.g.* higher modes (*c.f.* **Ba24**).

For mode 2 (bottom row in Figure 2), the patterns are more difficult to interpret. The generation in the vicinity of the continental slope is still visible, although the corresponding zone is more confined near the shelf. Loss of coherence can be identified slightly offshore of the generation patch (see at 65° W, 41° N). Again, the energy surface density reflects that the coherent part does not propagate far offshore, while the incoherent energy is more distributed in space. The alternating zones of positive and negative energy flux divergence is likely a signature of the similar magnitude of energy transfers between the

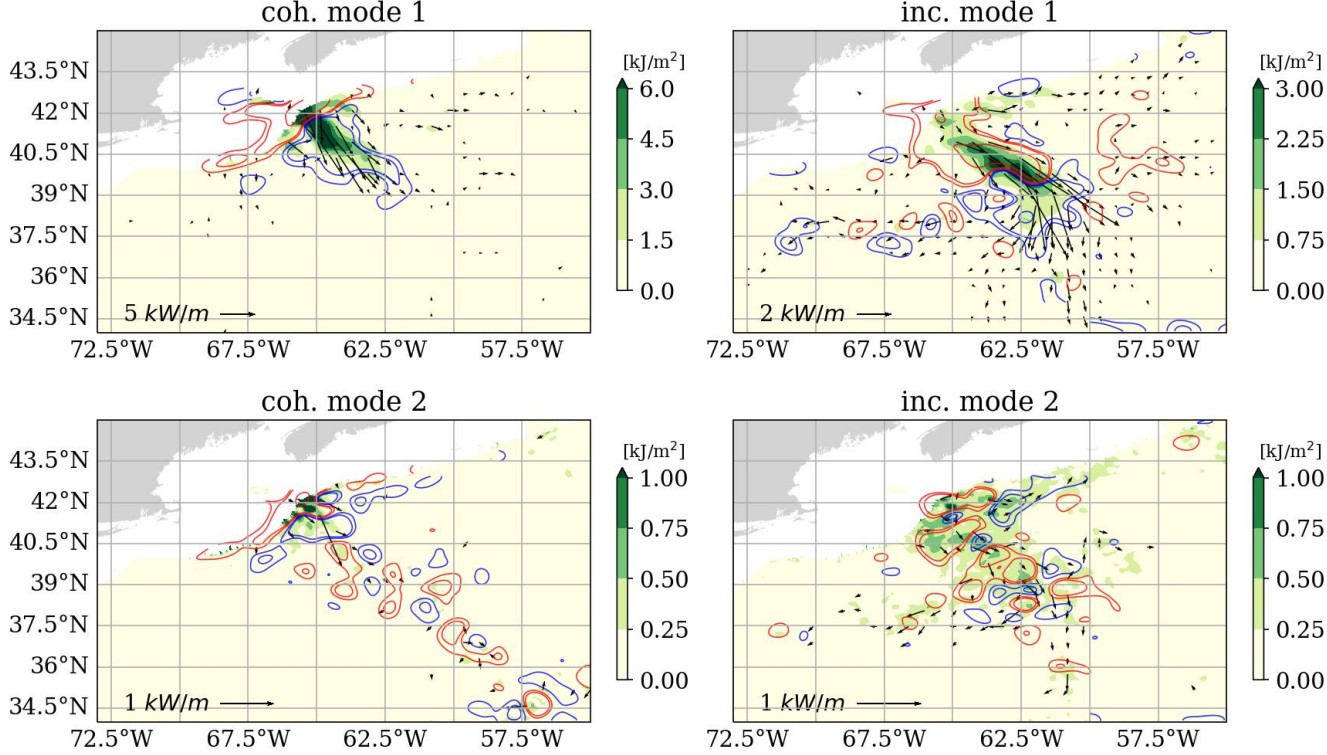

**Figure 2.** Internal tide energy in the Gulf Stream area: modal energy surface density (green shading), modal energy flux (arrows) and its divergence (smoothed, blue/red contours for negative/positive values, respectively) for the mode 1 (top) and 2 (bottom). Left panels show the coherent energy, right panels show the incoherent energy. Contours for the energy flux are 5, 1 and $0.5\,\mathrm{mW/m^2}$ for the coherent mode 1, 1.25, 0.25 and $0.13\,\mathrm{mW/m^2}$ for the incoherent mode 1 and 0.5, 0.01, $0.05\,\mathrm{mW/m^2}$ for the coherent and incoherent mode 2.

coherent and incoherent part of mode 2 on the one hand, and with other vertical modes on the other hand. Higher modes (not shown) exhibit much more complex patterns which are also difficult to interpret.

The same diagnostics performed at different months (not shown) exhibit qualitatively the same behaviour and yield to the same conclusions, although the precise location of the different patterns differ due to the variable path of the Gulf Stream. As a final remark, we notice that the tunnelling effect of the Gulf Stream, which describes deflection of the IT beam that initially arrives perpendicular to the main current and rotate and align to propagate upstream (Kelly and Lermusiaux, 2016; Duda et al., 2018), is visible in the incoherent energy flux of both modes 1 and 2.

### 3.2 Quantification of the coherent-incoherent IT energy transfers

The dominant terms of the mesoscale-induced internal tide loss of coherence are computed and integrated in space over the three subdomains of interest (Fig. 1), and averaged over the four months considered. These terms are the advection by the

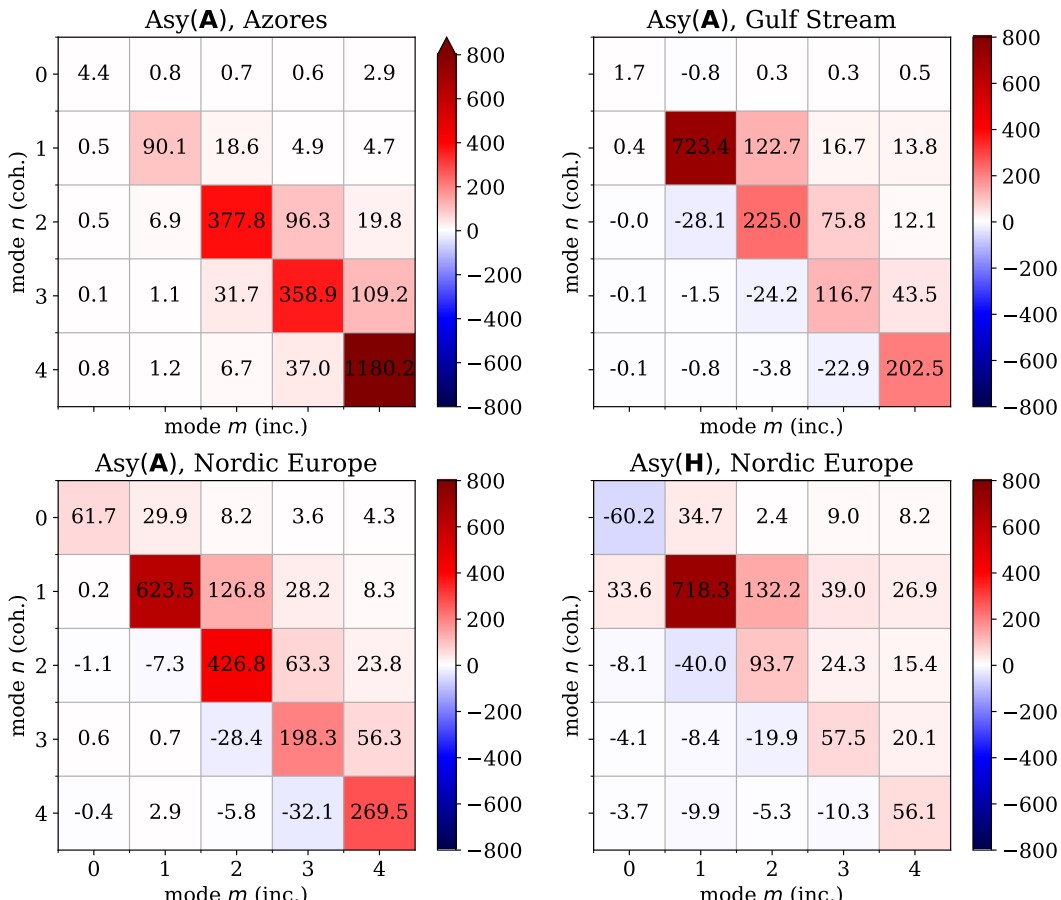

**Figure 3.** Antisymmetric part of the cross-modal coherent-incoherent interaction matrix associated with the advection in the Azores subdomain, Gulf Stream subdomain and Northern Europe, as well as the horizontal shear in the latter. It describes energy exchange from coherent mode number $n$ (row) to incoherent mode number $m$ (column). Positive values thus indicate a loss of energy for mode $n$, gain for mode $m$, *i.e.* a loss of coherence, and negative values indicate an opposite transfer. Values are in MW and are averaged over the 4 months considered.

background flow, in addition to the horizontal shear production term in the Northern Europe subdomain (shown Figure 3, coherent/incoherent and mode-wise antisymmetric part are shown).

We found that advection by the mean flow is a significant cause of loss of coherence, in all three subdomains, and is by far dominated by iso-modal interaction terms (diagonal in the plotted matrices). Horizontal shear is also important in the Northern Europe subdomain, and is also dominated by iso-modal transfers, although not as clearly as the advection. The vertical shear (not shown) is found to be negligible compared to the horizontal shear or advection, except for mode 1 in the Gulf Stream (iso-modal), which reaches 154 MW (compared to 723 MW for the advection term). These identified coherent-to-incoherent energy transfers are much larger than the corresponding symmetric terms (not shown), by one to two orders of magnitude, confirming that dissipative effects are not important in the loss of coherence. They are also greater – by at least a factor 2 –

than the corresponding cross-modal total (coherent + incoherent; not shown) energy transfers that were diagnosed in **Ba24**, and which were associated with exchanges from one mode to the next (higher) one. Conversely, the coherent-to-incoherent cross-modal energy transfers are smaller: for instance, in the Gulf Stream domain, from mode 1 and 2, there is $100\,\mathrm{MW}$ exchanged within the coherent IT, $197\,\mathrm{MW}$ within the incoherent IT (see Fig. A1); this reflects the incoherent energy fraction in this region shown in Fig. 1) while the coherent mode 1 gives $123\,\mathrm{MW}$ to the incoherent mode 2, and the flux from the incoherent mode 1 to the coherent mode 2 is small ($\approx 30\,\mathrm{MW}$; Fig. 3.b).

For comparison of the magnitude of the energy transfers, and for assessing the relevance of the diagnosed terms in the loss of coherence of the IT, we show the topographic scattering matrices for the coherent and incoherent IT in Figure 4. (Note that, by construction, the terms coupling coherent and incoherent IT vanish for the topographic scattering and the associated matrices are mode-wise antisymmetric.) We see that the incoherent barotropic modal conversion ($C_{0n}$, lower-left triangle in the plots) is nearly negligible, accounting for 5 to 10 % (in the Azores and the Gulf Stream, respectively) of the coherent barotropic modal conversion (upper-right triangles). Incoherent baroclinic scattering is not as weak, relatively to the energy level, which most likely reflects that the modal incoherent energy fraction is much higher for baroclinic modes than for the barotropic mode. Comparing Figure 3 and 4, one sees that the coherent-to-incoherent energy transfer (*e.g.* associated with advection by the mean flow) is a significant part of the barotropic conversion term in the Gulf Stream and Northern Europe subdomains. The relative importance of the advection-driven loss-of-coherence is much smaller in the Azores (less than 2% of the barotropic conversion for mode 1), which is in agreement with a weaker mesoscale activity. However, one may notice that the higher the mode, the greater the impact of the advection term, which contrasts with the other two subdomains. We do not have an explanation for this. As a result, the corresponding transfer for modes 2 and higher becomes non-negligible (for instance, it amounts 25% of barotropic conversion for mode 3): it is higher than the incoherent barotropic conversion term and of the same order of magnitude as the topographic scattering of the incoherent IT (see Figure 4).

### 3.3 Sensitivity to the vertical structure of the mesoscale flow in the Gulf Stream

The previous sections have shown that the mean flow can be very efficient to transfer energy between the coherent and incoherent internal tide while preserving the vertical mode involved. Amongst the various terms involved in this process in the coherent/incoherent modal energy equation, advection by the mean flow is dominant and significant in every subdomains considered. As a last investigation, we attempt to estimate which scales of the mean flow are involved in this coherent-to-incoherent energy transfer, focusing on the Gulf Stream subdomain. To this aim, we perform the vertical decomposition of the mean flow: $\overline{\boldsymbol{U}} = \sum_k \overline{\boldsymbol{U}}_k \phi_k$, in the $\overline{\boldsymbol{U}}_{nm}$ and $\overline{\boldsymbol{U}}^p_{nm}$ terms involved in $\boldsymbol{A}_{nm}$, *i.e.*:

$$\overline{\boldsymbol{U}}_{nm} = \sum_k \overline{\boldsymbol{U}}_k G^k_{nm}, \quad G^k_{nm} = \frac{1}{H} \int_{-H}^{\bar{\eta}} \phi_n \phi_m \phi_k \, dz, \quad \overline{\boldsymbol{U}}^p_{nm} = \sum_k \overline{\boldsymbol{U}}_k K^k_{nm}, \quad K^k_{nm} = \frac{1}{H} \int_{-H}^{\bar{\eta}} \frac{N^2}{c^2_m} \Phi_n \Phi_m \phi_k \, dz \, .$$

Here, $G^k_{nm}$ and $K^k_{nm}$ are modal interaction tensors, which give the amplitude of the advection-induced internal tide coupling, left aside the modal amplitude of the background flow and internal tide. This vertical mode decomposition of the mean flow allows to identify the vertical scales of the mean flow that most efficiently generate IT incoherence. It does not give readily

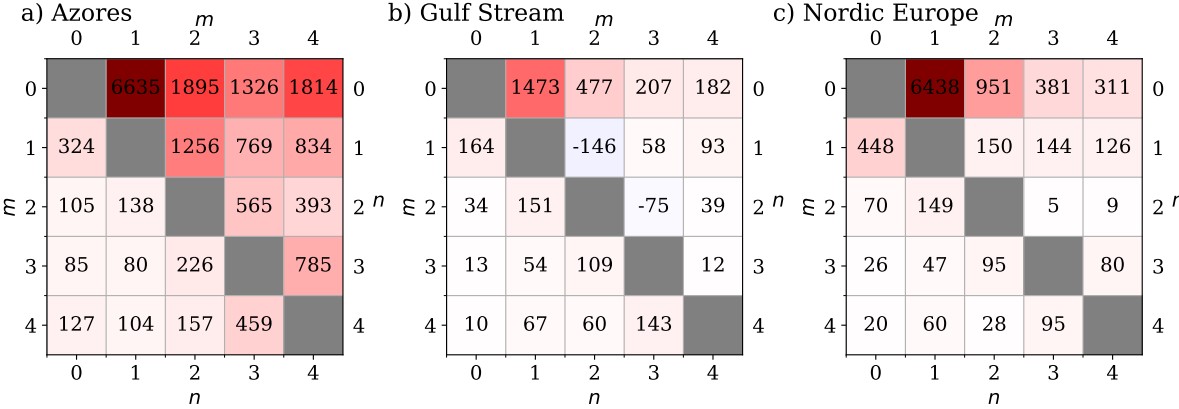

**Figure 4.** Antisymmetric part of the topographic scattering matrix in the Azores subdomain (a), Gulf Stream subdomain (b) and Northern Europe (c). Values are in MW. The coherent-coherent part ($C_{nm}^c$) is in the upper-right triangle, and the transposed incoherent-incoherent part is in the lower-left triangle ($C_{mn}^i$). Only half of each matrix is shown since they are antisymmetric. Sign convention is such that positive values indicate a loss of energy for the lowest mode number (column): $m$ in the upper-right triangle, $n$ in the lower-left triangle, and a gain for the higher mode number (row), *i.e.* an energy flux towards higher modes.

access to a separation of its horizontal scales (which was investigated by a different mean in Savage et al. (2020)) but still provide valuable information in this regard: for instance, the surface-intensified component of the flow is associated to smaller horizontal scales (*e.g.* compared to the barotropic component) and involves high mode numbers, as well as shorter time scales.

For simplicity, we restrict ourselves to the diagonal part in terms of IT vertical modes. We take the antisymmetric part, that is:

$$A_{kn}^{\text{asy}} = \left[ \mathcal{L}_c \left( \overline{U}_k \cdot \nabla v_n^i \right) \cdot v_n^c - \mathcal{L}_i \left( \overline{U}_k \cdot \nabla v_n^c \right) \cdot v_n^i \right] G_{nn}^k + \left[ \mathcal{L}_c \left( \overline{U}_k^p \cdot \nabla p_n^i \right) \cdot p_n^c - \mathcal{L}_i \left( \overline{U}_k^p \cdot \nabla p_n^c \right) \cdot p_n^i \right] K_{nn}^k / c_n^2.$$

Here, the average over the month of October is considered. This matrix, averaged over space, is shown in Figure 5, alongside the $G_{nn}^k$ modal interaction matrix and the standard deviation (with respect to time) of the modal amplitude of the mesoscale

currents. One sees that the mean flow is dominated by the first two modes (Fig. 5a). The interaction matrix $G_{nn}^k$ (Fig. 5b) is close to 1 for $k = 0$, as expected since $\phi_0 \approx 1$, and has finite amplitude for $k \approx n$, with reduced amplitude when $n > k$, all the more so for large mode numbers. Qualitatively, the matrix $K_{nn}^k$ exhibits the same features (not shown). All together, this results in the mode 0 and 1 from the mesoscale flow dominating, by large, the loss of coherence of the internal tide, as visible in Fig. 5c. We observe the same features in the Azores and Northern Europe domains (not shown), albeit only the mesoscale

mode 0 dominates the IT loss-of-coherence in the latter (its RMS amplitude is twice that of mode 1). An important remark is that the mesoscale mode 0 cannot trigger energy transfers between modes: indeed, since $\phi_0 \approx 1$, one recoves the orthogonality condition $G_{nm}^0 \approx K_{nm}^0 \approx \delta_{n,m}$. This explains, partially, why the loss of coherence is only marginally associated with exchange of energy between different modes.

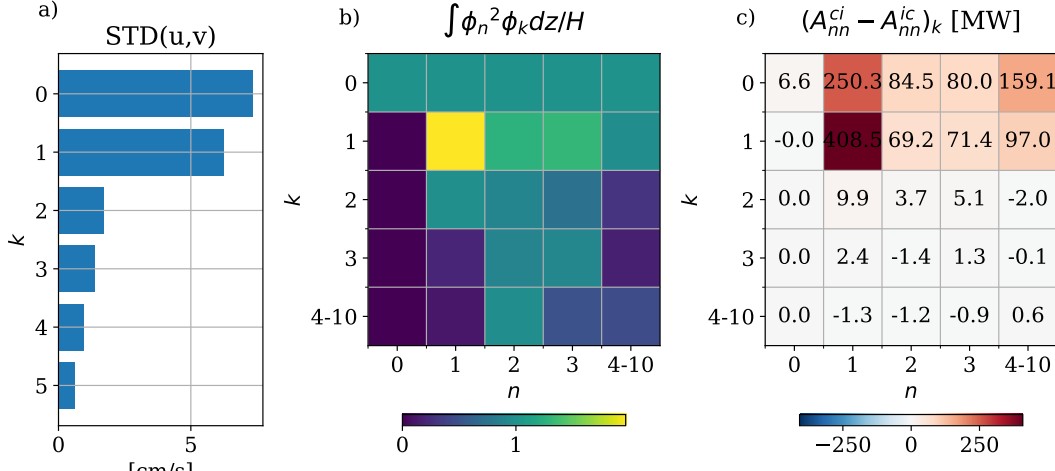

**Figure 5.** Decomposition of the loss of coherence induced by advection by the mean flow in the Gulf Stream subdomain (month of October): (a) modal space-averaged standard deviation (in time) for the low-frequency flow, (b) diagonal (IT mode wise) root mean square amplitude of the background advection interaction matrix $\int \phi_n^2 \phi_k$ and (c) domain-integrated mesoscale-mode decomposed diagonal antisymmetric coherent-to-incoherent interaction term. $k$ denotes the mesoscale mode, and $n$ the IT mode (panels b and c only)

Following these results, some remarks are worth mentioning. First, it indicates that realistic simulations are likely to be accurate in reproducing the IT loss of coherence, at least for the part driven by advection by the mean flow, since it is dominated by low baroclinic modes. Furthermore, and for the same reasons, it supports the validity of using reduced-order modelling of IT dynamics – e.g. for data assimilation – based on vertical mode projection of the primitive equations linearised around the background flow and truncated at a finite mode number (Kelly et al., 2021; Le Guillou et al., 2021).

### 3.4 Estimated scales of loss of coherence by the background flow

Let us summarise the results presented above and give a few "take-home numbers". These are given in Table 1 for the first three modes and correspond to the following quantities, averaged over each domain:

- topographic scattering (coherent + incoherent): $C_m = \sum_{n>m} C_{mn}$;

- advection by the mean flow (diagonal mode wise): $A_n = (A_{nn}^{ci} - A_{nn}^{ic})/2$;

- Horizontal shear $H_n$ and vertical shear $V_n$ terms, computed like the advection term.

A typical time scale is formed by dividing these terms by the corresponding mean modal energy, and a typical lengthscale by multiplying the latter by the mean modal group velocity. It represents the time (resp. length) necessary for the wave to lose coherence during its propagation. A short time (resp. short distance) is typical of large fluxes compared to the energy level.

The number obtained confirms that the loss of coherence is significant, especially in the Gulf Stream and Northern Europe domains, with typical transfer timescale of the order of 10 to 30 days (30 to 170 days in the Azores). The impact of vertical

| Domain | Mode | Topo. scat. MW | Topo. scat. loss time / length day / km | Adv. MW | Hor. shear MW | Vert. Shear MW | Coherence loss time / length day / km |
|---|---|---|---|---|---|---|---|
| Gulf Stream | 1 | 277 | 48 / 9400 | 723 | -11 | 154 | 15 / 3000 |
| | 2 | 133 | 41 / 4300 | 225 | -1 | 2 | 24 / 2540 |
| | 3 | 155 | 26 / 2000 | 117 | -1 | 2 | 34 / 2670 |
| North Europe | 1 | 677 | 35 / 3600 | 623 | 718 | 60 | 17 / 1740 |
| | 2 | 138 | 55 / 2500 | 427 | 94 | 3 | 14 / 660 |
| | 3 | 175 | 25 / 833 | 200 | 58 | 1 | 17 / 570 |
| Azores | 1 | 3180 | 6.7 / 1380 | 90 | -8 | 15 | 220 / 45500 |
| | 2 | 1342 | 10 / 930 | 380 | -4 | 6 | 35 / 3300 |
| | 3 | 1244 | 8 / 520 | 360 | -3 | 1 | 28 / 1820 |

**Table 1.** Modal and/or coherent-incoherent energy transfers in the three subdomains analysed and for the first three modes: topographic scattering (energy exchange with higher modes for the coherent+incoherent field) and associated time and length scales; coherent-to-incoherent iso-modal energy transfers due to the advection term, horizontal shear and vertical shear and associated (aggregated) typical time and length scale.

background shear is an order of magnitude weaker than advection by the background flow (slightly less in the Gulf Stream, slightly more in the Azores). The North East domain is particular in that the importance of the horizontal background shear is much stronger than in the other two domains, as was previously reported for the modal energy exchange of the total internal tide field in **Ba24**. The way in which the timescale and lengthscale evolve with mode numbers is intriguing. In the Gulf Stream, for example, the timescale increases with mode number, while the length scale remains almost constant. In the north-east domain, however, the timescale remains almost constant while the lengthscale decreases with mode number. In the Azores, both decrease with mode number. As discussed above, mode 1 exhibits a very slow loss of coherence. Further investigation is required to explain this behaviour, although it could be hypothesised that it is associated with differences in mesoscale variability (vertical mode and spectral energy content).

Globally, the numbers shown in Table 1 are consistent with the incoherent energy fraction based on a 1 month harmonic analysis shown in Figure 1 (right panels) and with the qualitative observations of the energy flux divergence (Figure 2). As was discussed in details in the previous subsection, it also reflects the intensity of mesoscale currents: averaged over the subdomain of interest, the mean RMS amplitude of horizontal currents for modes 0 to 5 is $10\,\mathrm{cm/s}$ in the Gulf Stream domain, $6,5\,\mathrm{cm/s}$ in the North Europe domain and $2,5\,\mathrm{cm/s}$ in the Azores domain. Once again, these numbers are based on a one-month coherent/incoherent separation. As mentioned in section 2, we also performed most of the analysis reported in this paper using a 3-month coherent/incoherent separation. The results obtained are qualitatively unchanged, except that the incoherent-to-coherent energy ratio as well as the coherent-to-incoherent energy transfers were found to be higher. This could be expected, for generally speaking, the longer the time window, the larger the incoherent-to-coherent energy ratio. Using a shorter or longer

time window would very likely lead to a change in the relative magnitude of the different processes. The impact of changing stratification would certainly increase with a longer time window, reflecting seasonality. Conversely, all transfer terms would decrease in magnitude with a shorter time window, reflecting the spectral content of mesoscale activity, which decreases as frequency increases. Such investigations are left for future work.

## 4 Summary and discussion

In this study, we examined the loss of coherence of the semi-diurnal internal tide induced by the mesoscale currents in the North-Atlantic using the NEMO-based realistic high-resolution numerical simulation eNATL60. We focused on three regions of interest (off the US North-East coast, around the Azores Island and in the North Eastern Atlantic) and on the loss of coherence that occurs over rather short timescales, wherein the coherent IT is defined over 1 month time windows. This analysis was based on a vertical mode decomposition of the hourly outputs of the simulation complemented by time-filtering to separate the internal tide from the background flow, allowing to estimate the transfer terms that appear in the corresponding coherent/incoherent modal energy budget formed from the linear coupled-mode equations for internal tides.

The main results of this study are as follows. First, coherent-to-incoherent energy transfers are significant in the IT energy budget and occur mostly without coupling between vertical modes. The corresponding energy transfer is found to be of the same order of magnitude as the barotropic conversion in the Gulf-Stream region. There, the typical energy transfer rates are of the order of a few 10 days / a few thousand of kilometres. Albeit of smaller importance, it is still not negligible in regions with weaker mesoscale activity such as around the Azores, especially for modes higher than 1. Second, loss of coherence is dominated by advection by the mean flow, although horizontal background shear is important in the Northernmost part of the domain. Vertical background shear is negligible, while the impact of the variability of the background stratification was not investigated. Finally, we found that the mesoscale modes 0 and 1 dominate by large this loss of coherence, which partially explains why loss of coherence mostly conserves the vertical mode of the internal tide.

It should be kept in mind than only a subset of the involved mechanisms has been investigated in this study – which is moreover based on a single numerical simulation. In particular, loss of coherence due to variations of the stratification (vertical and horizontal gradients) have not been addressed here. Furthermore, the effect of the variation of the basis of vertical modes in time, associated with such variations of background stratification, could obscure the conclusions, since all the investigated terms are basis-dependent. This is the main reason why we restricted ourselves to one-month period. Future work should address this limitation by combining the definition of the vertical mode basis with that of the separation into coherent and incoherent components. Such work would pave the way toward similar analyses over a longer time period, which are of particular interest as the cause of IT loss of coherence varies with the time scale considered (*e.g.* Buijsman et al., 2017; Zaron, 2022). This would allow a more complete and unified view of the processes underlying loss of coherence over different timescales to be built. It would also enable identifying the importance of the different causes – *e.g..* advection by the background flow vs. variation of the stratification (Savage et al., 2020; Zaron and Egbert, 2014) vs. incoherent generation (Bendinger et al., 2025) – for loss of coherence over different timescales, thus helping to refine simplified dynamical models of the internal tide dynamics as well

as identifying the key processes that are of importance for a realistic representation of the internal tides in realistic numerical simulations based on the primitive equations. Extension of the results presented in this paper over more regions of interest, in particular near the equatorial band, should also be addressed in future works.

The obtained results tend to confirm that loss of coherence can happen on fast time scales (shorter than a month), and is associated with energy transfers that are larger than between different vertical modes. In the context of internal tide mapping, *e.g.* from satellite altimeter data and especially in the context of the SWOT mission, these results tend to point at the need of designing inversion strategies that capture the incoherent fraction and the mechanism that are associated with this loss of coherence. Indeed, the typical revisit time of an altimeter satellite is around 10 days, during which the IT field can significantly decohere, thereby limiting the efficiency of time harmonic interpolation. At the same time, reduced-order modelling strategies based on vertical mode projections seem capable of capturing this process: indeed, coherent-to-incoherent exchanges are primarily iso-modal, and the role of mesoscale flow in this process is primarily played by low modes. The latter are easier to estimate than higher modes, which evolves faster. Consequently, low-mode truncation effects are likely to have a marginal impact, and simplified models of the internal tide could possess the necessary predictive capabilities to extract the internal tide signal from altimeter data.

*Code and data availability.* Material describing the NEMO ENATL60 simulation is available in Brodeau et al. (2020). The code used to perform the present analysis are based on the ITideNATL library https://github.com/NoeLahaye/ITideNATL, which can also be accessed in Lahaye (2024).

## Appendix A: Orthogonality of coherent and incoherent components

In this section, we explain why the coherent and incoherent components can be assumed to be orthogonal with respect to a time-averaging inner product. Savage et al. (2020) previously used this property, with a reference to Wunsch (2006) to justify it in cases where the coherent part is extracted by least-squares regression.

In theory, this property boils down to the orthogonality (in relation to the $L_2$ norm in time) of the Fourier constituents, for an infinite or periodic signal. In practice, to extract the coherent component, one has to minimize a squared residual on a finite-time window $[0, T]$: let us introduce a time series $y(t)$. Its coherent component $y_c(t) = \sum_k a_k e^{i\omega_k t}$ can be obtained by minimising the residual $\int_T |y - y_c|^2$ to find the $a_k$s, *i.e.* $\int_T y_c^*(y - y_c) = 0$. The last equality shows that the residual (the incoherent part) is orthogonal to the coherent part (with respect to the innerproduct associated with the time average over the considered time window).

This result can be extended to the case of two distinct fields, as long as the relative amplitude of the different harmonic constituents are the same. Let us consider a second time series $w(t)$, with a coherent component $w_c = \sum_k b_k e^{i\omega_k t}$ (with the same set of frequencies as above). Then, in order to get *i.e.* $\int_T w_c^*(y - y_c) = 0$, one must have $w_c \propto y_c$, *i.e.* $b_k = ba_k$ with $b$ a constant (conversely for $\int_T y_c^*(w - w_c) = 0$). This means that the relative weights of the frequency components are the same.

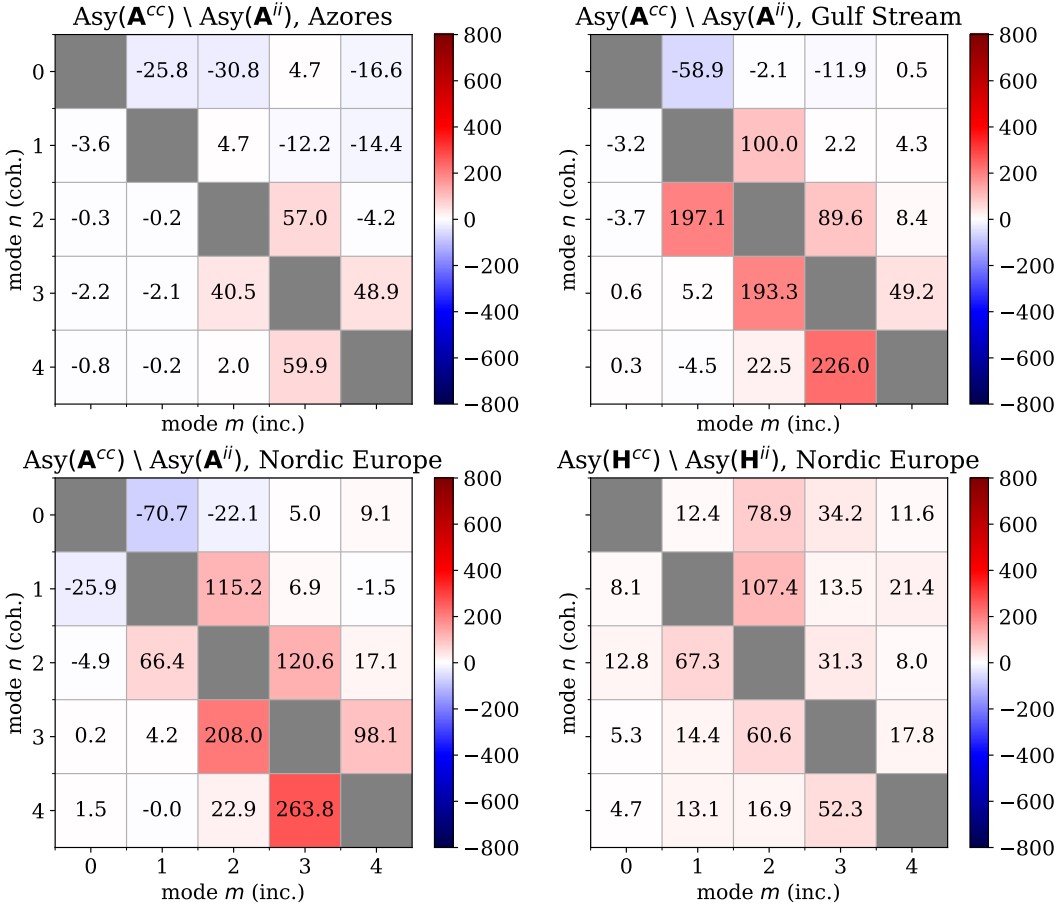

**Figure A1.** Antisymmetric part of the cross-modal interaction matrix associated with the advection in the Azores subdomain (a), Gulf Stream subdomain (b) and Northern Europe (c), as well as the horizontal shear in the latter (d). Values are in MW. What is shown is the coherent-coherent part int he upper triangle and transposed incoherent-incoherent part in the lower triangle. Same sign convention as in Fig. 4.

In the context of this paper, it seems reasonable to assume that the coherent part of $u$, $v$, $p$ have approximately the same relative frequency content, meaning that there coherent/incoherent part are almost orthogonal with each others. Numerical tests (not shown here) have confirmed that coherent/incoherent cross products neglected in the paper are indeed negligible.

## 435 Appendix B: Coherent / incoherent interaction terms

The figure A1 shows the antisymmetric part (mode-wise) of the coherent-coherent and incoherent-incoherent energy transfer terms associated with the advection by the mean flow and horizontal background shear (Northern Europe only for the latter). For the advection term coupling the coherent IT, this corresponds to: $(A_{nm}^{cc} - A_{mn}^{cc})/2$, and likewise for the incoherent IT and the horizontal shear.

*Author contributions.* AB designed and performed the data analysis and theoretical work, and led the interpretation of the scientific results. NL and GT participated in the conceptualization of the study and the interpretation of scientific results. AB and NL drafted the manuscript. All authors reviewed the manuscript and contributed to the writing and final editing.

*Competing interests.* The authors declare no competing interests to be reported

*Disclaimer.* TEXT

*Acknowledgements.* AB was partially funded by Région Bretagne through the ARED PhD funding program (COMIOE-project-15995). This research received support from the French research funding agency (ANR) under the ModITO project (ANR-22-CE01-000601), and from the CNES through the project DIEGOB. This work was supported by the French National program LEFE (Les Enveloppes Fluides et l'Environnement). The authors would like to thank the reviewers for their valuable feedback and questions.

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
