# Peer review of "Internal Tide loss of coherence in a realistic simulation of the North Atlantic"

_EGUsphere, 2025_

## Author Comment (AC1)

This is an important and timely study that provides a valuable quantitative analysis of internal tide (IT) incoherence. The theoretical framework is robust and the results are significant for the field. The following suggestions are intended to further strengthen the manuscript's clarity and impact.

Thank you very much for your positive feedback and useful suggestions. Our point-by-point response is below. The text will be modified accordingly as soon as the interactive discussion period is closed.

**Major comments**

**Expand on Physical Interpretation and Regional Differences.**

• The paper excellently identifies advection by the mean flow as the primary driver of decoherence. The discussion would be more powerful if it delved deeper into the physical reasons (line no. 232-233, 318-320) "why" this advection leads to a loss of phase relationship with the astronomical forcing.

The underlying dynamical process can be described as follows: at leading order, (i.e., provided the wave field can be described as a superposition of local plane waves), the advection term corresponds to the transport of the wave by the mean flow. This results in a local phase perturbation that propagates afterwards. As this process is not constant over time because the mesoscale flow evolves, the wave field becomes randomly perturbed and hence incoherent. Generally speaking, any term in the linearised equations that is not constant over time results in a loss of coherence, since, in the frequency domain, these terms are associated with triadic interactions between different frequencies. This introduces frequencies that are distinct from the initial coherent IT constituents.

• Furthermore, please expand the comparison of the three subdomains (Gulf Stream, Azores, Northern Europe). Explaining how the specific dynamics of each region (e.g., strength of jets, eddy activity) lead to the observed differences in coherence loss would significantly enhance the scientific narrative. (Line no. 308-310)

Following your suggestion and a comment from another reviewer, we will extend the description of the three different subdomains in the 'Data & Methods' section. Regarding the second part of your question, we will try to find a link, although this would probably require a dedicated study.

• Are the time-periods and areas analyzed in this manuscript representative of the broader North Atlantic?

The observed months cover late summer to winter. They should therefore be representative of the conditions one can encounter in the North Atlantic. The three areas chosen were also selected for their specific internal tides and mesoscale characteristics (as previously mentioned, we will expand on the description of these subdomains in the revised paper). Although this is based on our previous study, which did not specifically investigate the loss of coherence, these regions seem representative of the broader North Atlantic, with the obvious exception of the equatorial and Arctic zones. Extension of our analysis to a broader domain, including other ocean basins, would of course be valuable.

**Clarify Methodological Choices and Scope.**

The study's conclusions rely on key methodological choices that should be more thoroughly justified to ensure the results are robust and reproducible.

• Please state and justify the number of vertical modes used /resolved in the analysis. The paper notes that truncation error is negligible for modes below 5, but the total number used isn't specified. (Line no. 185)

Eleven modes were used in the present paper, ranging from n = 0 (barotropic tide) to n = 10 (internal tide). This number of modes is generally sufficient to describe most of the internal tide energy (see, for example, Falahat et al., 2014), particularly away from the generation site. Furthermore, the spatial resolution of the simulation does

not allow higher modes to be resolved everywhere in the basin. These details will be added to the text.

• The choice of a one-month window to separate coherent and incoherent tides is critical. The authors rightly note this is to avoid issues with a time-varying stratification basis, but a more detailed discussion of how a shorter or longer window might affect the results would be beneficial. (Line no. 308-310)

We will expand on the discussion of the results obtained with a three-month time window compared to the one-month analysis. In general, the longer the time window, the greater the level of incoherence and the greater the associated coherent-to-incoherent energy transfers (although we will not attempt speculating on the behaviour at an infinite time limit). More specifically, a longer time window captures more fluctuations of the advection term (which are contained in the mean for a shorter time window). This results in greater loss of coherence, associated with an increase in A (nm)^(ci).

• Briefly explain the rationale for the filter choices, such as the 2-day low-pass filter, to help readers understand their impact on the separation of the IT field and the mesoscale flow. (Line no. 188)

The two-day low-pass filter was chosen to retain as much mesoscale variability as possible while discarding the diurnal tide and most of the near-inertial waves (except near the equator --the near-inertial period is two days at a latitude of 14°). The complex demodulation period was chosen to fall in the middle of the three semi-diurnal tidal components. This will be specified in the text, alongside an expanded description of the internal tide signal extraction process.

**Strengthen the Link to Observations and Applications.**

The work is highly relevant to satellite altimetry, especially the SWOT mission. The paper would have greater impact if you explicitly discuss how these model findings can guide the interpretation or processing of real-world observational data. A discussion on how this analysis could improve the detection of incoherent internal tides in global SWOT data would be really useful.

We will expand upon the discussion in the final section. Indeed, our results suggest that over the time window corresponding to the typical revisit time of satellite altimeters (especially SWOT), loss of coherence is active and largely induced by interaction with the mean flow (more specifically, via the advection term). Our analysis suggests that intermodal coupling for the internal tide is not dominant in this process and that low-mode mesoscale modes are primarily involved. This suggests that simple models with low-mode truncation are capable of capturing the core dynamics.

**Future Directions.**

Suggest concrete next steps. For example, could this framework be applied to more regions, longer datasets, or models with different resolutions? What are the limitations/challenges if this work has to extended beyond the regions discussed here?

Following this suggestion and your previous comment, we will clarify the perspectives and indicate some remaining questions that should be answered. In particular, the role of varying stratification (and the resulting variation in the propagation speed of the IT modes) has been investigated previously in the literature (e.g. Savage et al 2020, Zaron \& Egbert 2014). Its role in this context could still be clarified and compared with the more direct effect of interaction with mesoscale dynamics. This will provide a more complete and robust view of IT dynamics and loss of coherence over sub-seasonal timescales.

**Minor comments**

• Figure Clarity. The interaction matrices are very informative, but their clarity could be improved. Please ensure the captions for Figures 3 and 4 explicitly define the sign convention to make them more immediately

understandable.

Will do, thanks.

**References**

- 1. Falahat, S., Nycander, J., Roquet, F. & Zarroug, M. Global Calculation of Tidal Energy Conversion into Vertical Normal Modes. J. Phys. Oceanogr. 44, 3225–3244 (2014).
- 2. Savage, A. C., Waterhouse, A. F. & Kelly, S. M. Internal tide nonstationarity and wave-mesoscale interactions in the Tasman Sea. J. Phys. Oceanogr. 1–52 (2020) doi:10.1175/JPO-D-19-0283.1.
- 3. Zaron, E. D. & Egbert, G. D. Time-Variable Refraction of the Internal Tide at the Hawaiian Ridge. Journal of Physical Oceanography 44, 538–557 (2014).

---

## Author Comment (AC2)

Esteemed Reviewer, We are appreciative of the constructive feedback provided on our paper. The ensuing responses are intended to address the comments that have been raised. Following the conclusion of the interactive discussion, a revised version of the paper will be made, incorporating the necessary modifications.

**Section 2.1: More information should be provided regarding the model setup. How long was the model run, and what is the spin-up time? Why this model was selected?**

We will expand upon the description of the numerical simulation in section 2.1. To answer your questions, the numerical simulation was run for 13 months, after an 18-month spin-up period during which tidal forcing was activated for the last six months. The simulation started from a 1/12° reanalysis (GLORYS12v1). This simulation was chosen because it is one of the few in the world to resolve a realistic internal tide field over an entire basin, and because the authors have access to the hourly outputs for post-processing.

**Model validation is an important part of the study. Please summarize the results from the referenced studies instead of expecting the reader to explore those papers independently.**

We will include the relevant information in the revision. In summary:

- -> A comparison of the barotropic tide with the FES2014 tidal atlas, which is used for boundary tidal forcing in the simulation, showed good agreement, particularly with regard to the dominant semi-diurnal amplitude.
- -> A comparison of the mesoscale field with the AVISO/DUACS product (i.e. comparing the standard deviation of daily-averaged SSH at a similar spatial resolution) shows a reasonable degree of agreement. However, eNATL60 is more energetic, which is to be expected given the coarser resolution of the AVISO/DUACS product. Further intercomparison of submesoscale-permitting numerical simulations by Ushida et al. shows that eNATL60 falls within the range of various models in terms of mesoscale energy and dominant patterns (e.g. the mean location of the Gulf Stream).
- -> A comparison of semidiurnal energy with drifter data, and of coherent IT (although the time window for its definition is inevitably different), shows fair agreement (Lahaye et al., 2025, SI). The main beams are captured in eNATL60 (compared to HRET), although the amplitude is larger, as expected given the shorter time window for computing the coherent signal using harmonic analysis. Conversely, the surface semidiurnal agrees with the drifter-derived estimate within a factor of 0.5 to 1.5 across most of the domain.

**Line 63: Explain "partial steps"**

'Z-coordinate with partial steps' means that the vertical levels are at a fixed, horizontally homogeneous depth, except in the vicinity of the seafloor, where the depth of the final level is equal to the depth of the seafloor (see Madec et al., 2019, Fig. 3.5). This information will be added in the revision.

**Line 74: Why were these four months specifically chosen?**

These months were chosen to provide an even coverage of the eight-month simulation period and to capture any seasonal variations that could be observed during this time.

**Line 75: What criteria were used to select these regions? Additionally, could you summarize the findings from Ba24 that influenced the choice of these regions?**

These regions were chosen because they exhibit three different configurations of the internal tide and its interactions with the low-frequency circulation, as previously demonstrated in Ba24. This can be summarised as follows: the Azores domain features strong IT generation that propagates through weak mesoscale currents; the Gulf Stream domain features IT generation and propagation inside an energetic mesoscale field; and the northern domain shows IT generation where the waves are unable to propagate very far, as well as active, slowly varying circulation. In Ba24, these three domains were found to be representative of three different dynamical regimes for the internal tide energy budget (5 other subdomains exhibited very similar properties): the Azores were dominated by interactions with the topography; the Gulf Stream region exhibited a forward energy cascade caused by the advection of the IT by the slowly varying circulation; and the northern domain

hosted energy exchanges between the slowly varying flow and the internal tide through shear production terms. This will be added to the revised paper.

**Line 125: Please explain the rationale behind the assumption that the coherent and incoherent components are orthogonal.**

In theory, and for an infinite or periodic signal, this properties is merely the orthogonality (with respect to the  $L_2\$  norm in time) of Fourier constituents. In practice, the extraction of the coherent component involves minimization of the squared residual  $\int |x-y-y|^2, i.e.$  i.e.  $\int |x-y-y|^2 + |x-y|^2 + |x$

Section 3.4: As Table 1 contains important information, please elaborate on it in this section. Discuss the physical factors in these regions that may be influencing the results. Additionally, mention if others have observed similar outcomes.

Thank you for your suggestion, we will expand the corresponding section in the revised version of the manuscript.

---

## Author Response (AR1)

Dear editors and reviewers,

we hereby resubmit our manuscript for publication in Ocean Science. We have addressed all the questions raised by the reviewers. Our point-by-point response can be found in our final response to the interactive process. A manuscript with changes highlighted in blue is also attached to this re-submission. We are grateful for your constructive remarks.

Sincerely,

the authors